# A framework for irrigation performance assessment using WaPOR data: the case of a sugarcane estate in Mozambique

**Abebe D. Chukalla**[1], **Marloes L. Mul**[1], **Pieter van der Zaag**[1,2], **Gerardo van Halsema**[3], **Evaristo Mubaya**[4], **Esperança Muchanga**[5], **Nadja den Besten**[6,2], **and Poolad Karimi**[1]

[1]Department of Land and Water Management, IHE Delft Institute for Water Education, 2611 AX Delft, the Netherlands CE1
[2]Water Management Department CE2, Delft University of Technology, 2600 AA Delft, the Netherlands
[3]Water Resources Management Group, Wageningen University & Research, 6700 AA Wageningen, the Netherlands
[4]Xinavane Estate, Xinavane, Mozambique
[5] CE3 Inclusive Agri-Food Value Chain Development Programme (PROCAVA) – FDA, Maputo, Mozambique
[6]VanderSat B. V., Agri, Food and Commodity Unit CE4, Wilhelminastraat 43a, 2011 VK Haarlem, the Netherlands

**Correspondence:** Abebe D. Chukalla (a.chukalla@un-ihe.org)

**Abstract.** TS1 CE5 The growing competition for finite land and water resources and the need to feed an ever-growing population require new techniques to monitor the performance of irrigation schemes and improve land and water productivity. Datasets from FAO's portal to monitor Water Productivity through Open access Remotely sensed derived data (WaPOR) are increasingly applied as a cost-effective means to support irrigation performance assessment and identify possible pathways for improvement. This study presents a framework that applies WaPOR data to assess irrigation performance indicators, including uniformity, equity, adequacy, and land and water productivity differentiated by irrigation method (furrow, sprinkler, and centre pivot) at the Xinavane sugarcane estate, Mozambique. The WaPOR data on water, land, and climate are in near-real time and spatially distributed, with the finest spatial resolution in the area of 100 m. The WaPOR data were first validated agronomically by examining the biomass response to water, and then the data were used to systematically analyse seasonal indicators for the period 2015 to 2018 on $\sim 8000$ ha. The WaPOR-based yield estimates were found to be comparable to the estate-measured yields with $\pm 20\%$ difference, a root mean square error of $19 \pm 2.5\,\mathrm{t\,ha^{-1}}$ and a mean absolute error of $15 \pm 1.6\,\mathrm{t\,ha^{-1}}$. A climate normalization factor that enables the spatial and temporal comparison of performance indicators are applied. The assessment highlights that in Xinavane no single irrigation method performs the best across all per-formance indicators. Centre pivot compared to sprinkler and furrow irrigation shows higher adequacy, equity, and land productivity but lower water productivity. The three irrigation methods have excellent uniformity ($\sim 94\%$) in the four seasons and acceptable adequacy for most periods of the season except in 2016, when a drought was observed. While this study is done for sugarcane in one irrigation scheme, the approach can be broadened to compare other crops across fields or irrigation schemes across Africa with diverse management units in the different agroclimatic zones within FAO WaPOR coverage. We conclude that the framework is useful for assessing irrigation performance using the WaPOR dataset.

## 1 Introduction

Increasing agricultural production to feed the growing global population can be achieved through either expanding agricultural land or by increasing productivity of the existing agricultural areas. With growing competition and scarcity of finite water and land resources, as well as the environmental and social costs of expanding agricultural land (Hess et al., 2016), improving irrigation performance indicators including land and water productivity has a clear preference.

The increasing global demand for sugar is also reflected in the steady increase in sugarcane production in Mozambique at an average annual rate of 10 % (FAO, 2019). The

majority of this increase comes from expanding agricultural land (Hess et al., 2016). Whilst Moraes et al. (2018) estimate there is a vast potential for expanding sugarcane production in Mozambique ($\sim$ 15 % of the land area is suitable for sugarcane production), water and land resources in the country are under increasing strain due to land degradation (Sutton et al., 2016), sectoral competition, and climate effects (e.g. drought and flood) (Van der Zaag and Carmo Vaz, 2003; Arndt et al., 2011). With the land productivity well below the global average (Binswanger-Mkhize and Savastano, 2017; Nkamleu, 2013), and amongst the lowest in the southern African region (Johnson et al., 2014), there is an opportunity to meet the demand without expanding the agricultural land. Thus, raising sugarcane productivity per unit of land and water on existing croplands needs to be explored by conducting irrigation performance assessment.

Monitoring irrigation performance indicators is key in checking general health, comparing the spatial and temporal performances of the scheme, and looking for causes and providing corrective action that aims at improving overall service provision and productivity (Molden et al., 1998; Bos et al., 2005). Traditional irrigation performance assessment considers indicators that can be categorized as (i) water balance, water service, and maintenance, (ii) environment, and (iii) economic indicators. The water balance, water service, and maintenance indicators are water fluxes CE6 and production-based indicators. The water delivery CE7 and production-based indicators include uniformity (evenness of water distribution within fields), equity (uniformity of water distribution between fields), adequacy (sufficiency of irrigation delivery compared to the requirement), land productivity (production per unit area), water productivity (production per unit water use), and efficiency (the fraction of productive water use) (Molden and Gates, 1990; Bos, 1997; Molden et al., 1998). These irrigation performance indicators were assessed using field data such as flow (discharge), crop yield, and plot level water consumption estimates using lysimeters or crop models (Araya et al., 2011; Dejen, 2015; Edreira et al., 2018).

Recent developments and improvements of remote-sensing (RS) products offer a viable alternative (Bastiaanssen et al., 1996; Karimi et al., 2011). RS-derived data have been increasingly applied as a cost-effective means for irrigation performance assessment. RS-derived irrigation performance assessment is based on production and actual water consumption, the latter of which is fairly CE8 considered the net outcome and result of effective rainfall and irrigation, allowing for a hydrological assessment and quantification of the net water abstracted by irrigated crops. In addition, it provides spatially distributed data, covers long periods and wide areas, and can be done retrospectively (Bastiaanssen et al., 1996; Karimi et al., 2011). Field data, in contrast, do not represent the spatial variation across an irrigation system well and are costly to obtain (Bastiaanssen et al., 2000). Traditional and RS-based performance assessments are complementary as the former has strength in observing the horizontal water fluxes such as discharges, while the latter has strength in observing high-resolution vertical water fluxes and biomass production.

Earlier studies provide insight into the application of RS-derived data to assess irrigation performance indicators. In this research, the earlier RS-based irrigation performance assessment studies are strengthened by considering a simple consistency check to validate the RS-derived data for established biomass response to water consumption (Steduto and Albrizio, 2005) and by introducing a comprehensive framework that guides the step-by-step translation of RS-derived datasets into irrigated agricultural performance indicators. In addition, the current study introduces a climate normalization factor that enables the spatial and seasonal comparison of irrigation performance indicators. The climate normalization factor is applied to distinguish climatic factors from agricultural management factors in their effect on irrigation performance.

This study first evaluates the FAO's portal to monitor Water Productivity through Open access Remotely sensed derived data (WaPOR) for consistency based on the established agronomic principle (biomass response to water consumption). It is then used to develop a framework to assess irrigation performance indicators, including adequacy, uniformity, equity, and land and water productivity. This framework is then used to assess the irrigation performance at Xinavane sugarcane estate differentiated by irrigation method.

## 2   Materials and methods

### 2.1   Study area

The study focusses on one of the largest sugarcane estates in Maputo province in Mozambique, the Xinavane estate. The estate is located on the banks of the Incomati River, approximately 136 km northwest of Maputo. This region is characterized by optimal conditions for sugarcane production in terms of climate, soils, and water availability. With a seasonal long-term average precipitation of 721 mm yr$^{-1}$ (den Besten et al., 2020), sugarcane production requires irrigation water, especially during the dry season, supplied by the Incomati River.

The most important water infrastructure in the Incomati Basin in Mozambique is the Corumana Dam, which was built for improving flood control and regulating downstream irrigation abstractions (including Xinavane) and hydropower production (de Boer and Droogers, 2016). Xinavane sugarcane estate, despite receiving allocations from the dam, remains largely vulnerable to climate variability. During a recent drought in 2016, reservoir levels in the Corumana Dam dropped drastically, and little water was available for irrigation in the Xinavane sugarcane estate. This resulted in a significant reduction in sugarcane production in 2016 compared

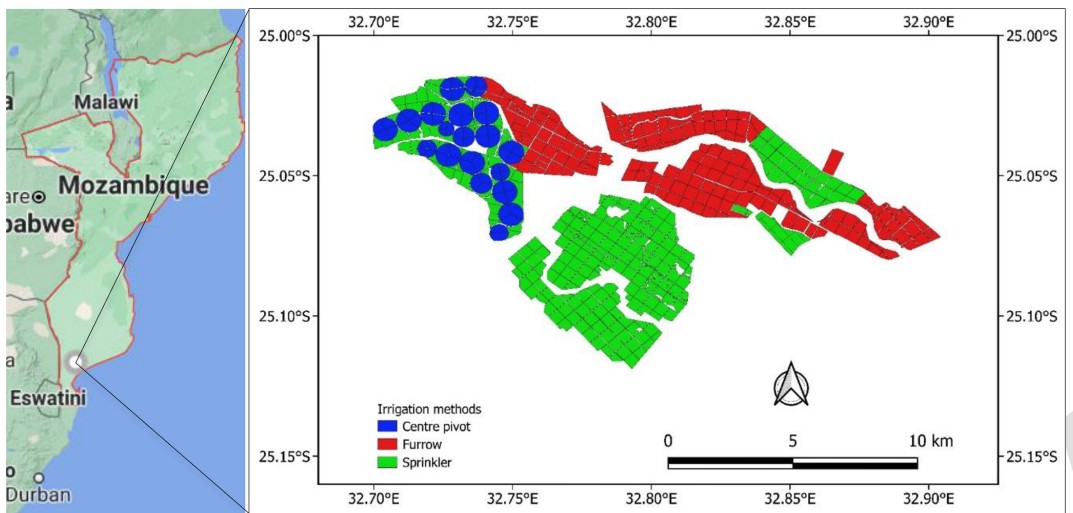

**Figure 1.** Irrigated areas (estate-operated) with different application methods at Xinavane sugarcane estate, Mozambique, shown on the larger-scale map of Mozambique (map data © 2021 Google, AfriGIS (Pty) Ltd.).

to previous years (Tongaat Hullet, 2018). Such events are expected to continue to occur. To partially address this, Mozambique put drought mitigation measures in place for the Xinavane area, including the construction of the new Moamba Major Dam ($760 \times 10^6$ m³ TS2) and the heightening of the Corumana Dam wall, which will result in a capacity increase from $879 \times 10^6$ m³ to $1260 \times 10^6$ m³ (Tongaat Hullet, 2018).

The widely used irrigation methods at the Xinavane sugarcane estate are furrow, overhead sprinkler (hereinafter referred to as sprinkler), and centre pivot irrigation (Fig. 1). A total of 8027 ha categorized into 387 georeferenced fields and three irrigation application methods are considered in our analysis. Furrow, sprinkler, and centre pivot irrigation covers 3343, 3629, and 1055 ha, respectively. The average field size under furrow, sprinkler, and centre pivot irrigation methods is 17, 18.3, and 55.8 ha, respectively. All fields in the sample are operated and managed by the estate; fields operated by outgrowers were excluded from the analyses.

## 2.2 WaPOR datasets

TS3 Datasets from FAO's portal to monitor Water Productivity through Open access Remotely sensed derived data (WaPOR; URL: https://wapor.apps.fao.org/home/WAPOR_2/1, TS4) are used for the analyses as it provides the required layers to estimate both land and water productivity. The database covers Africa and the Near East regions in near-real time for the period from 2009 to date (2021) (FAO, 2020c). WaPOR datasets are available at the continental scale (Level 1 at 250 m), country scale (Level 2 at 100 m), and project level (Level 3 at 30 m). The latest WaPOR version (WaPOR v2.1) is an improvement from WaPOR v1.0, following the quality assessments by IHE Delft and ITC (Mul and Bastiaanssen, 2019; FAO, 2020a). The methodology

used for compiling the actual evapotranspiration of WaPOR is based on the ETLook method (Bastiaanssen et al., 2012) and further developed by the FRAME consortium (the full description of the methodology is provided in FAO, 2020b). WaPOR v2.1 was found suitable for inter-plot comparison of irrigation performance indicators for plots larger than 2 ha (Blatchford et al., 2020).

At Xinavane, the finest resolution of the WaPOR data is 100 m (Level 2). The WaPOR Level 2 datasets used in this study include layers for actual evaporation ($E$), transpiration ($T$), and net primary production (NPP) at a dekadal (10 d) timescale. In addition, daily precipitation at 5 km resolution, daily reference evapotranspiration at 20 km resolution, and annual land cover classification (LCC) at 100 m resolution were used. The precipitation ($P$) and reference evapotranspiration (RET) datasets were resampled to 100 m resolution using the nearest-neighbour resampling techniques (GDAL, 2021). An overview of the WaPOR data used in the analyses is presented in Table 1.

Although there is a continuous WaPOR L2 dataset (100 m) available from 2009 to date (2021), only the data from 2014 are derived that stems CE9 from the PROBA-V satellite. The data prior to 2014 are derived from resampled L1 (250 m) data, which are obtained from the MODIS satellite. Since this creates a discontinuity in the data as observed by Chukalla et al. (2020b), the pre-2014 data have been discarded in this analysis, and only data starting from the 2014–2015 growing season onwards have been selected.

## 2.3 A framework for assessing irrigation performance using WaPOR data

Figure 2 shows the flowchart describing the approach to assessing WaPOR-based irrigation performance indicators at

**Table 1.** The WaPOR layers used for the analyses.

| WaPOR layer | Spatial resolution | Temporal resolution (coverage) |
|---|---|---|
| Evaporation ($E$) | 100 m | |
| Transpiration ($T$) | 100 m | |
| Net primary production (NPP) | 100 m | Dekadal (2014–2018) `TS5` |
| Precipitation ($P$) | 5 km | |
| Reference evapotranspiration (RET) | 20 km | |
| Land cover map (LCC) | 100 `TS6` | |

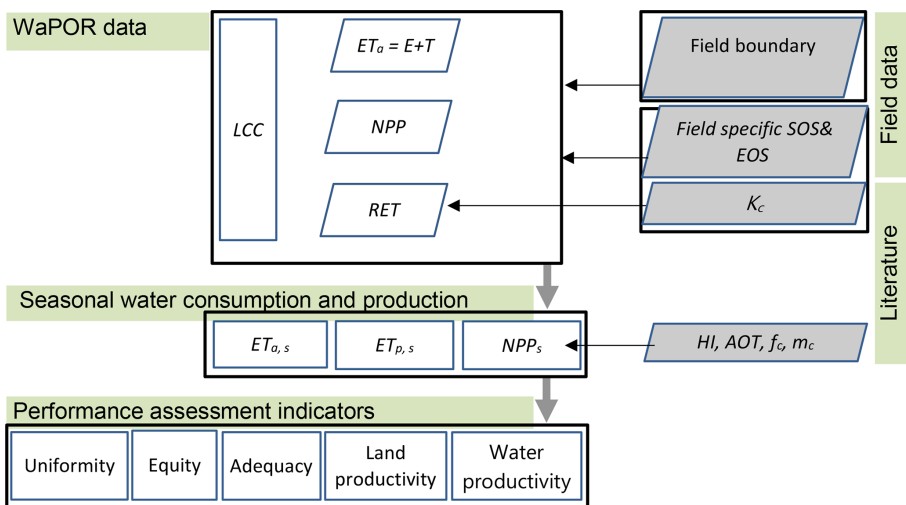

**Figure 2.** Schematic representation of WaPOR-based irrigation performance assessment framework.

the Xinavane sugarcane estate. Irrigation performance indicators are derived from WaPOR and field data in three main steps. First, actual evapotranspiration ($\mathrm{ET_a} = E + T$), reference evapotranspiration (RET), and net primary pro-
5 duction (NPP) layers of FAO WaPOR are preprocessed to match the spatial resolution, remove non-crop pixels using crop map or land cover classification (LCC), and undergo a quality check. Second, the seasonal $\mathrm{ET_a}$ ($\mathrm{ET_{a,s}}$), seasonal potential evapotranspiration ($\mathrm{ET_{p,s}}$), and seasonal NPP ($\mathrm{NPP_s}$)
10 are calculated from their respective WaPOR layers between the start of the season (SOS) and end of the season (EOS) for each plot. $\mathrm{ET_{p,s}}$ is derived from RET and crop coefficient ($K_c$). Finally, the irrigation performance indicators are analysed. At this stage, $\mathrm{NPP_s}$ is translated to above-ground
15 biomass (hereafter referred to as biomass – $B$) using crop-specific information – above over total biomass (AOT) for non-root corps or below over total for root and tuber crops, light use efficiency correction factor ($f_c$), and moisture content of fresh biomass ($m_c$). The biomass is multiplied by har-
20 vest index (HI) to derive the crop yield. The remainder of this section describes the input data and equations used in each step in more detail.

### 2.3.1 Seasonal water consumption and crop yield

**Growing season**

The sugarcane estate operates on a ratooning system. Thus, 25 the start of the growing season (1 d after harvesting) and end of season (next year's harvesting date) vary per field. The actual growing period of each field was used to calculate the production per unit of land and per unit of water consumed. The average length of the growing season is $347 \pm 32$ d. This 30 study covers four growing seasons: season 1 (2014/15), season 2 (2015/16), season 3 (2016/17), and season 4 (2017/18), reported as 2015, 2016, 2017, and 2018, respectively, i.e. the year the fields are harvested (Fig. 3).

**Seasonal water consumption** 35

Actual water consumption refers to the amount of water that is depleted from the root zone through the process of transpiration by a crop and direct evaporation from the soil represented by WaPOR $E + T$ ($\mathrm{ET_a}$). The seasonal $\mathrm{ET_a}$ is the total actual water consumption during the cropping season. 40

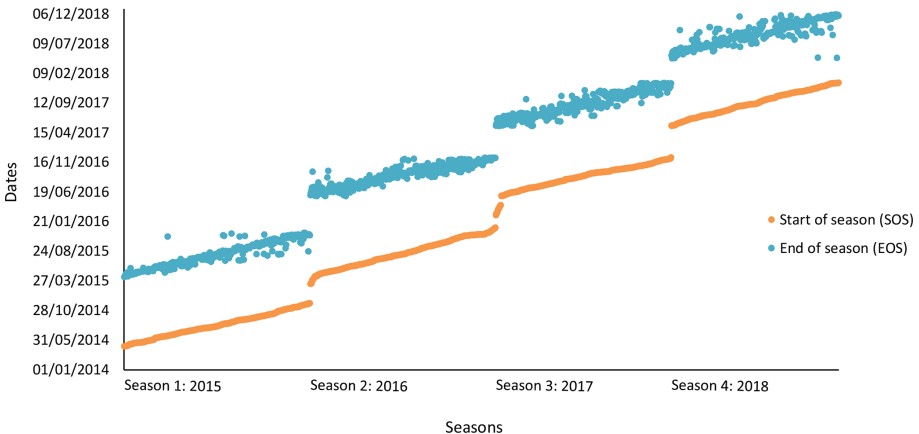

**Figure 3.** The start and end of season for individual fields for the four growing seasons at Xinavane estate.

## Crop yield

The season NPP layer from WaPOR, accumulated over the crop-growing period (Fig. 3), is converted to above-ground biomass ($B$) in kilograms per hectare ($kg\,ha^{-1}$) and crop yield ($Y$) in kilograms per hectare using Eqs. (1) and (2) (Mul and Bastiaanssen, 2019):

$$B = \text{AOT} \cdot f_c \cdot \frac{\text{NPP} \cdot 22.222}{(1 - m_c)}, \tag{1}$$

where $m_c$ [–] is the moisture content of the fresh biomass, $f_c$ [–] is the light use efficiency (LUE) correction factor calculated by dividing the LUE of the crop (in this case sugarcane) by the LUE of a generic crop type that WaPOR NPP layer uses ($2.7\,MJ\,g^{-1}$ biomass; FAO, 2018, 2020b), and AOT [–] is the ratio of above-ground over total biomass. $B$ and $Y$ can be expressed in tonnes per hectare ($t\,ha^{-1}$), by dividing the amount in kilograms per hectare by 1000. Crop yield is calculated by multiplying the biomass by the harvest index (HI [–]):

$$Y = B \cdot \text{HI}. \tag{2}$$

In absence of field data, literature was consulted to estimate these crop parameters. Table 2 presents the values and the source of the parameters.

The WaPOR-based sugar cane yield was validated with sugarcane yields as measured by the Xinavane estate for four seasons on 387 fields. In addition, the WaPOR-based biomass and water consumption were checked for consistency with agronomic principles. An increasingly strong linear relationship is expected between biomass and evapotranspiration (Steduto and Albrizio, 2005), between biomass and transpiration (De Wit, 1958), and between biomass and normalized transpiration (Steduto and Albrizio, 2005), whereby the normalized transpiration is the sum of the daily ratio of transpiration over reference evapotranspiration over the crop season (Steduto et al., 2007).

### 2.3.2 Performance assessment indicators

The irrigation performance indicators selected for this study are uniformity, equity, adequacy, and productivity; these were selected as these could be assessed (sometimes with a slight modification) using the WaPOR data. These performance indicators are further explained below, and the set of equations for water-consumption-based performance indicators are presented in Table A1.

Uniformity measures the evenness of water consumption within an irrigated field. It is calculated by assessing the coefficients of variation (CV) of seasonal $ET_a$ within a field. Thus, uniformity is 1 minus the CV (Ascough and Kiker, 2002). It serves as a measure for the heterogeneity of soil water storage capacity and thus water storage efficiency in a field. It can serve as a proxy for irrigation distribution uniformity (Burt et al., 1997) in farms where the management is central and the same level of inputs is consistently applied (e.g. variable rate input application in not CE10 practices). Other factors like soil type, fertility, pest, and crop variety can also affect actual water consumption and thus uniformity. Thus, the CV of seasonal $ET_a$ indicates the combined effect of all factors (water, fertility, pests, diseases, and salinity).

According to Pitts et al. (1996), the acceptable standard uniformity of irrigation application distribution for centre pivot, sprinkler, drip, and furrow irrigation methods is 75 %, 75 %, 85 %, and 65 %, respectively. The distribution uniformity exceeding the standard threshold is considered excellent.

Equity measures the evenness of water consumption between fields within an irrigation scheme with a homogenous crop, which could be a proxy for an even distribution of water to the different irrigated fields. It is calculated as the CV of the average ET of each field, which is an indication of equity in the scheme. A CV of 0 % to 10 % is defined as good equity, CV of 10 % to 25 % as fair equity, and CV > 25 % as poor equity (Bastiaanssen et al., 1996; Karimi et al., 2019).

**Table 2.** Parameters used in the biomass and yield analyses of sugarcane.

| Parameter | Description | Value | Source |
|---|---|---|---|
| $m_c$ | Moisture content of fresh crop biomass | 59 % | Yilma (2017), Mul and Bastiaanssen (2019) |
| $f_c$ | Light use efficiency correction factor | 1.6 | Villalobos and Fereres (2016 TS7) |
| AOT | Ratio of above-ground over total biomass (AOT) | 1 | FAO (2020c) |
| HI | Harvest index | 1 | FAO (2020c) |

Adequacy ($A$) is the measure of the degree of agreement between the actual water use and crop water requirement (Bastiaanssen and Bos, 1999; Clemmens and Molden, 2007). Adequacy is estimated as the ratio of seasonal actual evapotranspiration ($ET_{a,s}$) over seasonal potential evapotranspiration ($ET_{p,s}$) (Kharrou et al., 2013; Karimi et al., 2019). Potential evapotranspiration is the maximum crop evapotranspiration from disease-free and well-fertilized cropped fields under optimum soil water conditions; it is calculated by multiplying reference evapotranspiration by the crop coefficient (Allen et al., 1998). The seasonal $ET_{p,s}$ of sugarcane is aggregated from the monthly value of crop coefficient multiplied by the reference evapotranspiration (Table A2). Good adequacy performance is defined for the range of $0.8 < A \leq 1$ TS8, acceptable range $0.68 < A \leq 0.8$ TS9, and poor performance $A \leq 0.68$ TS10 (Karimi et al., 2019).

Productivity is a measure of benefit generated per unit of resource used. The benefit could be biophysical, economic, and/or social; the resource base could be consumed or supplied water or land covered by the crop (Zwart and Bastiaanssen, 2004; Hellegers et al., 2009; Karimi et al., 2011). This study focussed on biophysical production per unit of land or evapotranspiration, also known as land and water productivity.

Land productivity is defined as biomass production or crop yield per unit of land. For water, we similarly distinguish biomass water productivity ($WP_b$) and crop yield water productivity (WP). $WP_b$ is defined as the ratio of biomass over seasonal $ET_{a,s}$, whereas WP is defined as the yield over $ET_{a,s}$. Since for sugarcane we use a harvest index of 1, $WP_b$ is equal to WP here.

Spatial–temporal variations can be caused by both management practices and climate. Figure B1 shows a correlation between water productivity and reference evapotranspiration ($r^2$ of 0.5, 0.7, and 0.8 for furrow-, sprinkler-, and centre-pivot-irrigated fields, respectively). The correlation between actual evapotranspiration and reference evapotranspiration (Fig. B2) is even stronger ($r^2 > 0.8$). Thus, to exclude the climate-related factor, we normalized the water productivity and evapotranspiration using a climate normalization factor. This is defined as the ratio of the weighted average reference evapotranspiration (weighted based on the field size and growing length of the fields) to the reference evapotranspiration at the field (Eq. 3).

$$f_{norm} = \left( \frac{\overline{RET}}{RET_i} \right), \tag{3}$$

where $f_{norm}$ [–] is the normalizing factor for the selected indicator, $\overline{RET}$ is weighted average reference evapotranspiration, and $RET_i$ is reference evapotranspiration at a field in millimetres per season.

## 2.4 Consistency check of WaPOR data

Figure 4 shows the relationship between biomass ($B$; WaPOR-derived and WaPOR-observed) and water consumption of irrigated fields categorized by irrigation method for the year 2018 (with the Supplement, Fig. S1, showing the other 3 years from 2015 to 2017). In furrow- and sprinkler-irrigated fields, the WaPOR-derived biomass and actual evapotranspiration show a high correlation (a minimum $r^2$ of $\sim 0.83$ ($n \approx 150$) in 2015, 2017, and 2018 and $r^2 \approx 0.63$ in the relatively dry year of 2016), indicating consistency between the two independently generated datasets. For the centre-pivot-irrigated fields, $r^2$ is much lower, with a value of $\approx 0.6$ in 2015, 2016, and 2017 and the lowest $r^2$ of 0.2 ($n \approx 19$) in 2018. The low number of fields irrigated by centre pivots may have contributed to the low correlation. Moreover, the estate-observed yield at Xinavane sugar estate versus $ET_a$ shows a high spread and thus a low correlation ($r^2 \approx 0.13$).

Table S1 in the Supplement provide the analyses of the relationship between biomass and transpiration and biomass and normalized transpiration for the entire period of analyses (2015–2018). In contrast to expectations based on agronomic principles, the correlation decreases when considering biomass and transpiration ($\sim 0.80$) and biomass and normalized transpiration ($\sum T_a/\overline{RET}$) ($\sim 0.71$) (see further CE11 Supplement). The accuracy of the evaporation and transpiration split in WaPOR is therefore questioned; this was also observed by Mul and Bastiaanssen (2019). Further analyses will therefore only focus on indicators that use evapotranspiration, not evaporation and transpiration, as input. For instance, the beneficial fraction (i.e. the ratio of transpiration over evapotranspiration) is not included in the analysis. Yet, two tests based on WaPOR-derived biomass and total actual evapotranspiration ($ET_a$) have confirmed the agronomic expectations (Table S2). The first is that the correlation coefficient of the linear regression line passing through the origin for the biomass vs. normalized actual water consumption is

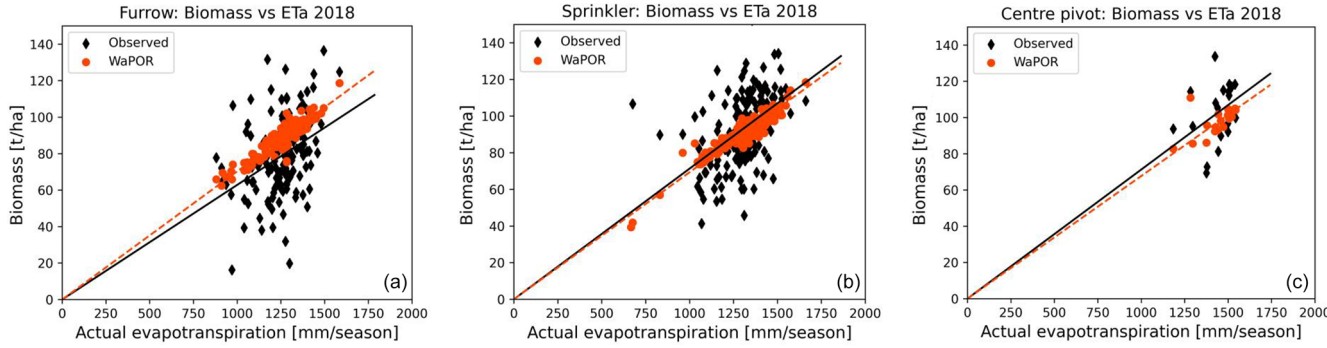

**Figure 4.** The relationship between biomass (as measured by the estate and derived from WaPOR) and actual evapotranspiration (derived from WaPOR) of furrow- **(a)**, sprinkler- **(b)** and centre-pivot-irrigated **(c)** fields at Xinavane sugar estate, harvested in 2018.

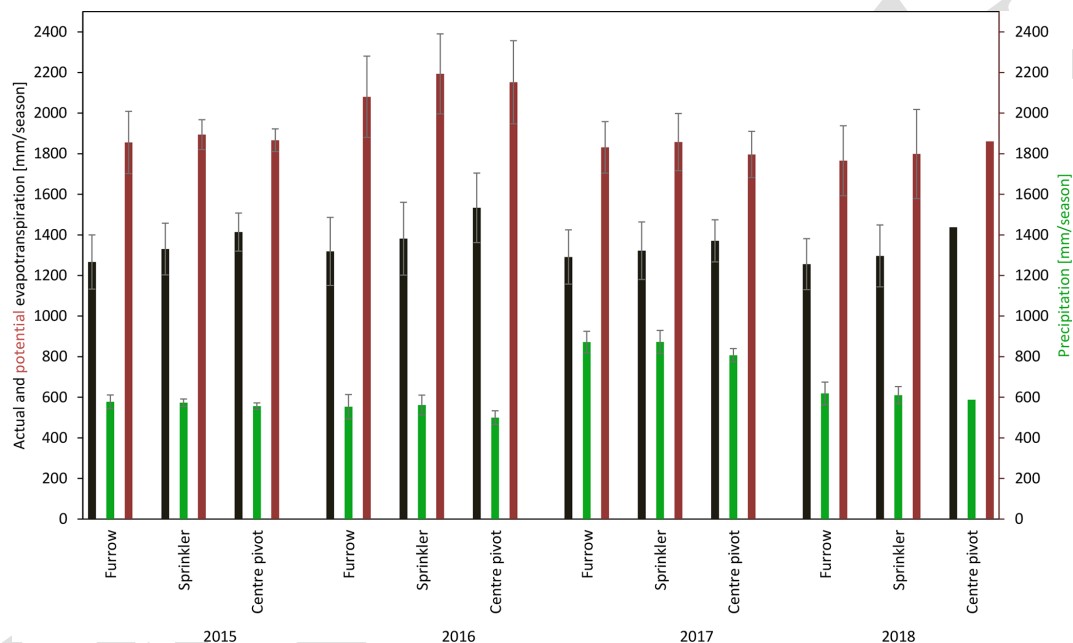

**Figure 5.** Seasonal actual and potential evapotranspiration and precipitation at Xinavane sugar estate from 2015 to 2018. The error bar indicates the variation across the fields irrigated by different methods.

higher than that of the correlation coefficient for the biomass vs. actual water consumption. Second, the crop water productivity normalized by reference evapotranspiration (WP*) is confirmed to be conservative and within the range of values for $C_4$ crops (30–35 $g\,m^2$), including sugarcane (Steduto et al., 2007, 2009).

## 3 Results

### 3.1 Seasonal water consumption

Figure 5 shows the seasonal actual and potential evapotranspiration and seasonal precipitation at Xinavane sugarcane estate, distinguished by the three irrigation application methods. The four-season (2015 to 2018) average precipitation is 640 mm per season and ranges from the minimum of 500 mm per season in 2016 to the maximum precipitation of 875 mm per season in 2017. The four-season average $ET_a$ at Xinavane is 1350 mm per season, and its average seasonal values range between 1255 mm per season in 2018 at furrow-irrigated fields to 1533 mm per season in 2016 at fields irrigated using centre pivots. In the four seasons the $ET_a$ is significantly the highest ($P$ value $< 0.05$) at fields irrigated using centre pivots, followed by sprinklers and furrows (Table A4).

The high *average* $ET_a$ over Xinavane irrigation scheme in 2016 coincides with the reported drought year. This mainly manifested itself with high $ET_{pot}$ as the annual precipitation that falls within the command area was not much lower than in 2015 and 2018. After normalization for climate variation,

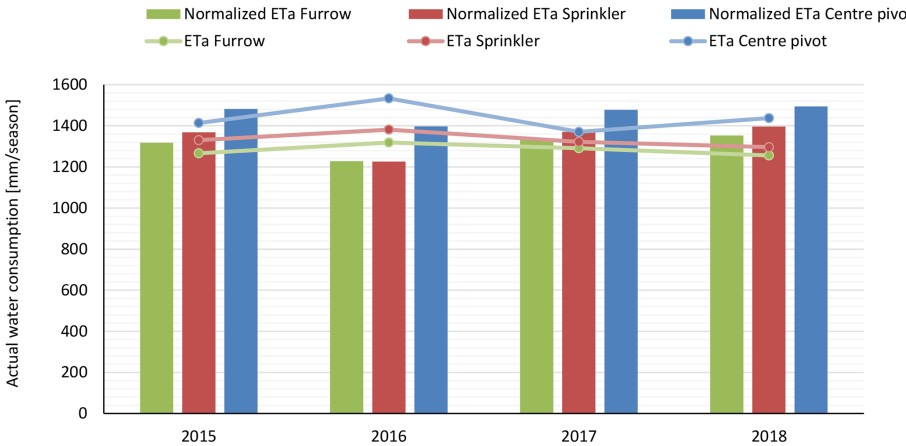

**Figure 6.** Normalized actual evapotranspiration at Xinavane sugar estate categorized by irrigation method from 2015 to 2018.

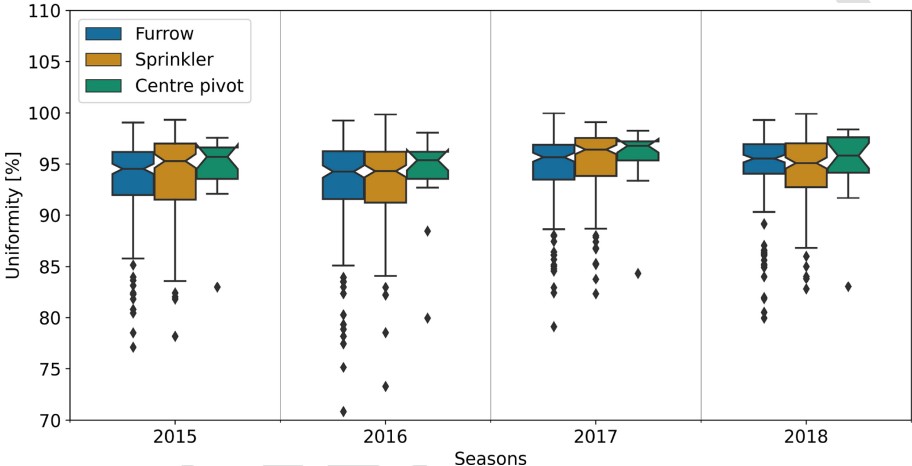

**Figure 7.** Coefficient of variation of actual water consumption per pixel inside a field at Xinavane sugar estate categorized by irrigation method from 2015 to 2018. The lower and upper whiskers in the box plot show the minimum and maximum values. The lower, middle, and upper bar of the box show the 25th, 50th, and 75th percentiles of the values.

the normalized $ET_a$ is actually lowest for 2016, indicating higher water deficit (lowest actual per unit of potential evapotranspiration), with the drought having more impact on sprinkler and furrow irrigation than on centre pivot irrigation. Despite the $ET_a$ being the highest in 2016, when normalized by climate, the results show that 2016 experiences the highest water deficit. The four-season average actual water consumption of centre pivots remains the highest followed by sprinklers and furrows, except for 2016, when the sprinkler-normalized $ET_a$ is at the same level as furrow $ET_a$ (Fig. 6). This indicates that the sprinkler system was more affected by the drought conditions in 2016 compared to the other systems.

## 3.2 Performance of irrigation delivery

### 3.2.1 Uniformity

The uniformity of water consumption within the fields is $\sim$ 94 % for all three irrigation methods (Fig. 7). The calculated uniformity is above the standard values per irrigation method and is therefore considered excellent. Centre pivots show an even higher uniformity than the other irrigation methods.

### 3.2.2 Equity

The average seasonal coefficient of variation (CV) of $ET_{a,s}$ among fields irrigated by the same irrigation method is 15 % (Fig. 8). Fields irrigated using furrows, with a CV of 18 %, have the highest heterogeneity in water consumption compared to areas irrigated using the sprinkler (CV = 14 %) and centre pivot irrigation method (CV = 13 %). The coefficient

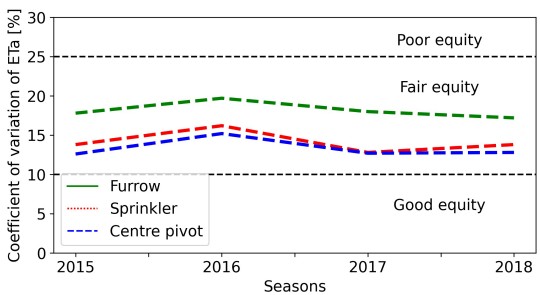

**Figure 8.** Coefficient of variation of actual water consumption between fields irrigated by different methods at Xinavane sugar estate from 2015 to 2018.

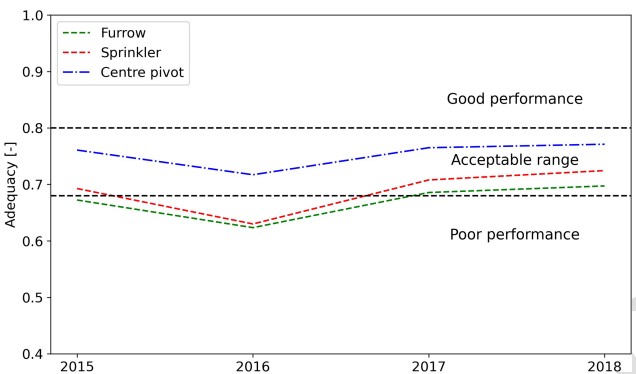

**Figure 9.** Adequacy [–] at Xinavane sugar estate categorized by irrigation method.

of variation of water consumption between fields irrigated by a particular irrigation method and thus equity of water use among the fields is considered fair.

### 3.2.3 Adequacy

The four-season average adequacy varies spatially across the Xinavane irrigation scheme, with visible differences between fields irrigated using centre pivots compared to fields irrigated using furrows and sprinklers for the period analysed. Figure 9 shows the highest adequacy for fields irrigated using centre pivots (0.75) followed by fields irrigated using sprinklers and furrows ($\sim$ 0.69). In the study period, the adequacy performance at fields under centre pivots falls in the acceptable range (from 0.68 and 0.8) for sugarcane (Karimi et al., 2019). The adequacy in fields under sprinkler and furrow irrigation also is acceptable except in the year 2016, which is recognized as a drought year, when adequacy was poor.

### 3.2.4 Land productivity

The 4-year seasonal average WaPOR-based yield is 89 t ha$^{-1}$ (86 t ha$^{-1}$ for fields irrigated using furrows, 88 t ha$^{-1}$ for areas irrigated using sprinklers, and 93 t ha$^{-1}$ for fields irrigated using centre pivots). For all years (except 2017), the highest sugarcane yield (land productivity) at Xinavane

is found in fields irrigated by centre pivots, followed by fields irrigated by sprinkler and furrow irrigation methods (Fig. 10).

The 4-year seasonal WaPOR yield is in the same order of magnitude compared to the estate-measured sugarcane yield: 86 t ha$^{-1}$ vs. 81.4 t ha$^{-1}$, 88 t ha$^{-1}$ vs. 93 t ha$^{-1}$, and 93 t ha$^{-1}$ vs. 99 t ha$^{-1}$ for fields irrigated using furrow, sprinkler, and centre pivot irrigation methods, respectively. Part of the minor discrepancy between the WaPOR- and estate-measured yield could be due to the selection of crop parameters such as harvest index and moisture content. Yet, the comparison between both yields shows acceptable statistics (Table A3 in the Appendix), with a root mean square error of 19 ± 2.5 t ha$^{-1}$ and mean absolute error of 15 ± 1.6 t ha$^{-1}$.

Whilst the average values for WaPOR-based yields are of the same magnitude as the estate-observed data (65 % of yield differences at the fields are within ±20 %), WaPOR overestimates relatively low yields (marks on scatter plot above 1 : 1 line) and underestimates relatively high yields (marks on scatter plot below 1 : 1 line) (Fig. 11). WaPOR yields thus show a marked less variation in yields than reported by the estate.

### 3.2.5 Water productivity

The seasonal and four-season average water productivity at Xinavane is shown in Fig. 12. The four-season average water productivity is the highest for furrow-irrigated fields (6.9 kg m$^{-3}$), compared to the values for fields irrigated with sprinklers (6.7 kg m$^{-3}$) and centre pivots (6.6 kg m$^{-3}$). One of the reasons for such differences is the fraction of ET$_a$ being utilized for productive purposes (transpiration) compared to non-productive evaporation. Raes et al. (2013 TS11) report that centre pivot and sprinkler irrigation wets 100 % of the field compared to furrow irrigation that wets $\sim$ 80 % of the field and thus results in higher evaporation rates, which is in line with our observations.

The large variation of WP over the years (Fig. 12) is also apparent after normalization for climate variation (Fig. 13). The normalized WP is highest in a relatively dry year (2016) compared to the other 3 years; this is opposite to WP, where 2016 has the lowest WP. It indicates that climate-related parameters expressed through potential evapotranspiration have a large impact on the WP. The normalized WP shows the variations which are related to management practices; during the drought of 2016, the Xinavane estate practised deficit irrigation, which is reflected in the high normalized WP values.

## 4 Discussion

### 4.1 The framework

The presented framework was used to conduct an irrigation performance assessment using WaPOR data. Our analysis shows that fields irrigated using centre pivots have the high-

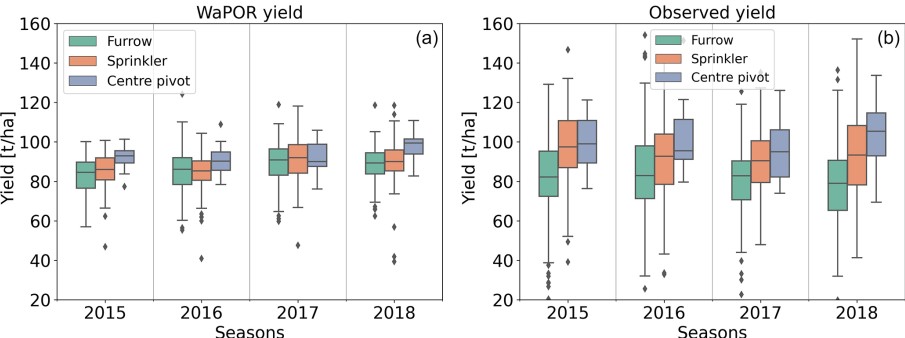

**Figure 10.** Box plot of yield at Xinavane sugar estate categorized by irrigation method from 2015 to 2018: WaPOR yield **(a)** and estate-measured (observed) yield **(b)**. The lower and upper whisker in the box plot show the minimum and maximum values across the fields irrigated by different methods. The lower, middle, and upper bar of the box show the 25th, 50th, and 75th percentiles of the values across the fields irrigated by different methods.

est equity, adequacy, and land productivity, followed by fields irrigated using sprinklers and furrows. This outcome agrees with the conclusion by Karimi et al. (2019), who assessed performance of irrigated sugarcane in Eswatini (Swaziland) by differentiating areas according to management regimes including irrigation methods. The adequacy performance under the three irrigation methods was generally acceptable, except in 2016, when performance of all three irrigation methods was poor. Fields under centre pivots do, however, have the lowest water productivity, followed by sprinkler and furrow irrigation, which is contrary to the finding by Karimi et al. (2019), who reported the WP of centre pivots to exceed that of furrow irrigation. In fact, it is claimed that pressurized irrigation (sprinklers and centre pivots) improves uniform distribution and application efficiency of irrigation water and increases crop yield (Magwenzi and Nkambule, 2003; Playán and Mateos, 2006). Yet, these irrigation methods increase seasonal evaporation (Playán and Mateos, 2006), which could be due to differences in percentage of land wetted. Our findings show that the uniformity of water consumption on the fields under the three irrigation methods is reasonably comparable and high ($\sim 94\%$), which can be regarded as excellent according to the standard set by Pitts et al. (1996). The high uniformity of water consumption in furrow-irrigated fields is in the same range as that of centre pivots and sprinklers, which is unlike what was found in South Africa (Griffiths and Lecler, 2001).

The results of normalization for climate differences of the water consumption and water productivity allows for comparison of the results under different climate conditions (different years). While the ranking for the different irrigation technologies according to the indicators remains the same, it clearly shows the impact of the climate. In particular, during the drought year of 2016 when the potential evapotranspiration was relatively high, the normalized water consumption was low, indicating higher water deficit compared to the other years. The impact on sprinkler-irrigated fields was the high-

est. On the other hand, the normalized WP during 2016 was the highest of all the years, even though the WP was lowest for the same biomass in 2016, indicating the climate having a large impact on non-beneficial evaporation.

This finding seems to suggest that production constraints can be addressed by taking certain measures, including improved farm practices. However, one factor that influences crop yield but that is difficult to influence, and that has not been assessed by this study, is the age of the crop. It is known that the early ratoons (harvests after first planting the cane) achieve significantly higher yields than subsequent ratoons (Mehareb and Galal, 2017). So, achieving the 90th percentile targets may not be easy for fields with older crops, even though the Xinavane Estate CE12 uses a higher target yield than the 90th percentile crop yield.

This study shows that the presented framework offers a systematic approach to assess irrigation performance indicators using WaPOR and field data. Five WaPOR-derived irrigation performance indicators, namely uniformity, equity, adequacy, and land and water productivity, are used to monitor the quality of the irrigation and agronomic services. Our framework builds on earlier studies that assess irrigation performance indicators based on RS (Karimi et al., 2019; Blatchford et al., 2020) and provides a comprehensive and simple step-by-step framework to conduct an agronomic evaluation using WaPOR data. The approaches in the framework are scripted with Python in Jupyter Notebooks that can be run on local machines, and Google Colaboratory (Colab) is published together with observed yield data in GitHub (Chukalla et al., 2020a). It shows that with limited field information (crop type and cropping season) and some parameters obtained from the literature, the analyses can be implemented.

### 4.1.1 Limitations of the WaPOR database

The linear relationship between the independently derived WaPOR biomass and water consumption agrees with the

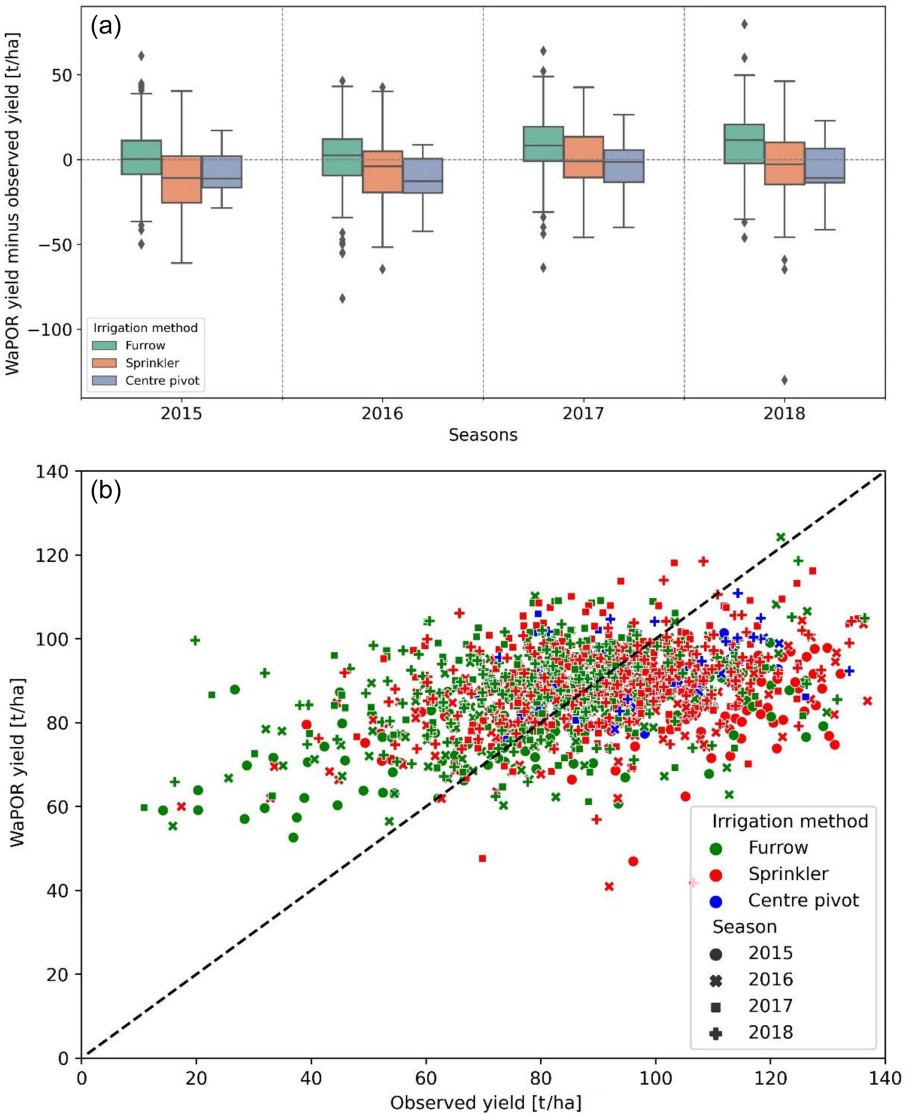

**Figure 11.** WaPOR yield compared to estate-observed yield: **(a)** the difference between estate-measured and WaPOR yield and **(b)** scatter plot of WaPOR yield vs. estate-measured yield.

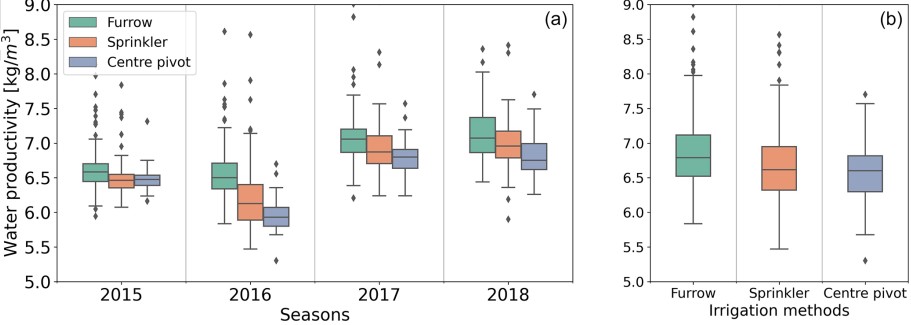

**Figure 12.** Box plot of water productivity in kilograms per cubic metre ($kg\,m^{-3}$) at Xinavane sugarcane estate categorized by **(a)** irrigation methods in 2015 to 2018 and **(b)** the four-season average. The lower and upper whiskers in the box plot show the minimum and maximum values across the fields irrigated by different methods. The lower, middle, and upper bars of the box show the 25th, 50th, and 75th percentiles of the values across the fields irrigated by different methods.

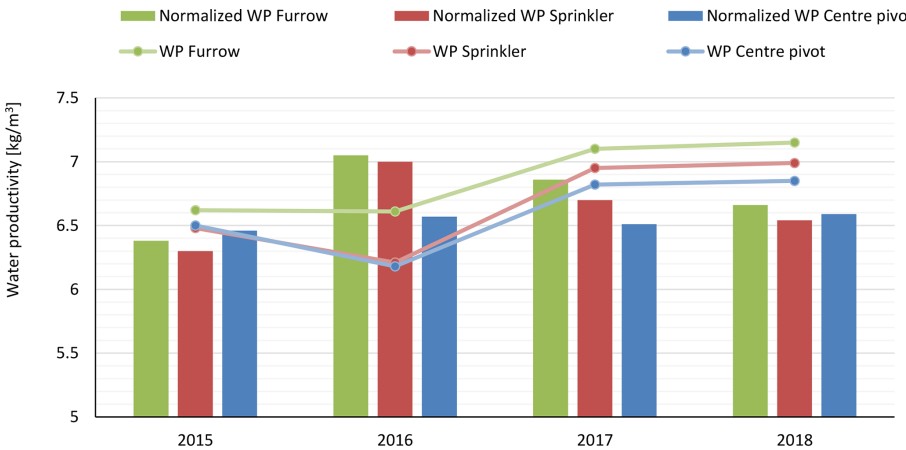

**Figure 13.** Normalized water productivity at Xinavane sugarcane estate categorized by irrigation method in 2015 to 2018.

expected agronomic principles (De Wit, 1958; Steduto and Albrizio, 2005). However, the correlation coefficient of the biomass versus actual evapotranspiration is higher than the correlation coefficient of the biomass versus transpiration and biomass versus normalized transpiration. This implies an inaccurate estimation of transpiration ($T$) and evaporation ($E$) in WaPOR. WaPOR separates the available energy into $T$ and $E$ using a factor $\alpha \cdot \mathrm{LAI}$, where $\alpha$ is the light extinction factor (FAO, 2018; Mul and Bastiaanssen, 2019). A review on values for $\alpha$ shows large differences between different land use classes and within land use classes (Zhang et al., 2016). Thus, WaPOR applying only one fixed value for $\alpha$ could have serious implications for the use of the $T$ and $E$ layers of WaPOR, such as in quantifying beneficial fraction (the ratio of transpiration over evapotranspiration).

Even though the analyses seem to be consistent with the understanding of how the different irrigation technologies perform, there are some known limitations of RS and WaPOR data in particular, which need to be mentioned here. These may stem from (i) the Land Surface Temperature (LST) data used by WaPOR (which are taken from MODIS and have a resolution of 1 km, used to derive moisture stress and thus to calculate the actual evapotranspiration and net primary production; this could be the cause for the reduced variation of WaPOR biomass data and may affect the spatial variation of evapotranspiration as well); (ii) land cover noise of non-sugarcane land use such as farm roads and irrigation and drainage infrastructures within a pixel; (iii) the number of cloud-free RS images on which the analysis and numerical interpolation are based (the fewer the cloud-free images, the poorer the data quality and the higher the uncertainty in the indicators one can expect); (iv) the time of day when the images are taken (determinant for which part of the daily ET curve is monitored and the time of day the water stress is more or less severe); and (v) the angle of image capture and its correction function.

The methods used in WaPOR for data production and statistical methods for the reconstruction of missing values are, however, on a par with those used in other RS-based products for monitoring agro-hydrological parameters developed by the scientific community. As such, some of these limitations are inherent to the use of remote sensing in general. Yet, our analysis shows consistency between the different datasets.

### 4.1.2 Limitation of the crop-related information

Crop-specific parameters such as harvest index, the moisture content of the fresh yield, and the ratio of above-ground over total biomass were fixed values and determined using literature and fieldwork in Ethiopia. However, it is known that these crop parameters can vary significantly based on climatic or field management conditions. Other variations may stem from differential exposure to pests and diseases and soil and rooting conditions caused by waterlogging (den Besten et al., 2021) and soil salinity, which are not catered for. We were unable to determine how much these assumptions affect the results. All these factors are potential sources of (slight) deviations in the numerical output of WaPOR that may lead to over- and underestimations of crop yield and WP.

Having noted this, we did perform a validation of the WaPOR biomass data using observed harvested cane data of more than 300 fields over four seasons. WaPOR biomass data for $\sim 65\%$ of the field level comparison differed within a $\pm 20\%$ range. The comparison between the estate-measured yield and WaPOR biomass showed acceptable statistics (Table A3).

### 4.2 The way forward

Investments in high-quality public domain global and regional remote-sensing data products for water and lands, us CE13 e.g. WaPOR datasets, have made it possible to conduct spatio-temporal analysis of irrigation performance at multiple scales from an irrigation scheme to district scale,

basin scale, and the whole country. This provides a great advantage, especially in areas where both water and land resources are scarce and in situ data are scant. This study presents a RS-based assessment framework and showcases the power of using the WaPOR dataset in providing spatial and temporal irrigation performance indicators. Such information cannot be generated with the data collected traditionally (point data) or would come at a significant cost.

Yet, accurate interpretation of the results, diagnosis of the causes of the performance variation, and formulation of practical solutions cannot be done unless the WaPOR analyses and results are complemented with observed data of field conditions (e.g. the level of water and nutrient inputs, waterlogging, and salinity levels) that can help explore the constraints. Though this limitation puts a disclaimer on our findings, the procedures in this study can provide a useful reference for similar future studies.

Subsequent studies could additionally consider socioeconomic performance indicators, such as social water productivity (e.g. employment per unit water or land use) and economic water productivity (economic return per unit water or land use), which could help to implement comprehensive performance assessment of irrigation schemes.

## 5    Conclusions

Remote-sensing datasets are increasingly applied as an innovative tool for monitoring the performance of irrigation schemes in order to improve land and water productivity amid the growing competition for finite and even dwindling resources (land and water). In this study, first, the remotely sensed FAO WaPOR dataset was successfully validated by comparing WaPOR-derived sugarcane yield with field observations, as well as agronomically CE14. The yield response to water confirms agronomic expectations: (i) the correlation between biomass and actual water consumption normalized for climate is stronger than the correlation between biomass and actual water consumption, and (ii) the water productivity of sugarcane normalized by reference evapotranspiration falls within the conservative values reported for $C_4$ crops. Second, the WaPOR-derived datasets were applied to assess irrigation performance indicators, including uniformity, equity, adequacy, and land and water productivity at Xinavane sugarcane estate, segmented by irrigation method. We conclude that the systematic approach demonstrated in the current study can serve as a framework to operationalize the use of WaPOR-derived data and other increasingly available RS-derived products for irrigation performance monitoring and assessment.

The comprehensive WaPOR-based irrigation performance assessment in this sugarcane state finds that fields irrigated by centre pivots have the highest adequacy, land productivity, and equity, followed by sprinkler- and furrow-irrigated fields, but the lowest water productivity.

We identified that part of the spatial and seasonal variation of indicators, water productivity, and seasonal water consumption in particular is explained by non-climatic factors that can be influenced by management interventions. Investigating the root causes of the land productivity variation and whether proper management of inputs and controlling of salinity and drainage could improve productivity and the overall performance require further study, including field-based observations.

## Appendix A: Tables

**Table A1.** Water-consumption-based irrigation performance assessment criteria and indicators.

| Criteria | Indicator | Equation* | Reference |
|---|---|---|---|
| Uniformity | CV of ET | CV of seasonal average $ET_a$ per pixel in a field | Karimi et al. (2019) TS12 |
| Equity | CV of ET | CV of seasonal average $ET_a$ per field inside the scheme/block | Karimi et al. (2019) TS13) |
| Adequacy | The ratio of $ET_{a,s}$ over $ET_{a,p}$ or relative evapotranspiration (RET) | $RET = \frac{ET_{a,s}}{ET_{p,s}}$ <br><br> $ET_{a,s} = \sum_{SOS}^{EOS} ET_a$ <br><br> $ET_{p,s} = \sum_{SOS}^{EOS} ET_{p,m}$ <br><br> $ET_{p,m} = \sum_{SOS}^{EOS} k_{c,m} \cdot RET_m$ | Karimi et al. (2019) TS14 |
| Land productivity | Biomass production ($B$) | $B = AOT \cdot f_c \cdot \frac{NPP_s \cdot 22.222}{(1-MC)}$ <br> AOT is above over total biomass, $f_c$ is light use efficiency correction factor, and MC is moisture content in fresh biomass. | Mul and Bastiaanssen (2019) |
| | Yield | $Yield = B \cdot HI$ <br> HI is harvest index. | |
| Water productivity | Biomass WP ($WP_b$) | $WP_b = \frac{B}{ET_{a,s}}$ | FAO 66 |
| | Crop yield WP (WP) | $WP = \frac{Y}{ET_{a,s}}$ | |

* where SOS and EOS are the start of season and end of season, $ET_{a,s}$ is seasonal actual evapotranspiration, $ET_{p,s}$ and $ET_{p,m}$ are seasonal and monthly potential evapotranspiration, $RET_m$ is monthly reference evapotranspiration, $k_{c,m}$ is the crop coefficient, and $NPP_s$ is seasonal net primary production.

**Table A2.** Crop coefficients of sugarcane.

| Crop stages | Duration of crop development stages | | $K_c$ values [–] |
|---|---|---|---|
| | Default in CROPWAT 8.0 (Smith, 1992) [d] | % | |
| Initial | 30 | 8 | 0.4 |
| Development | 60 | 16 | [0.4–1.25] |
| Mid-season | 180 | 49 | 1.25 |
| Late season | 95 | 26 | [1.25–0.75] |
| | 365 | | |

**Table A3.** Statistical comparison of WaPOR yield and estate-measured yield.

| Season | Irrigation method | Number of fields compared ($n$) | Root mean square error [t ha$^{-1}$] | Mean absolute error [t ha$^{-1}$] |
|---|---|---|---|---|
| 2015 | Furrow | 175 | 18.5 | 14 |
| ($n = 352$) | Centre pivot | 16 | 14.7 | 13 |
| | Sprinkler | 160 | 22.5 | 18 |
| 2016 | Furrow | 153 | 20.3 | 15 |
| ($n = 351$) | Centre pivot | 17 | 16.7 | 13 |
| | Sprinkler | 180 | 19.6 | 15 |
| 2017 | Furrow | 152 | 21 | 16.5 |
| ($n = 332$) | Centre pivot | 19 | 16 | 13 |
| | Sprinkler | 161 | 17 | 14 |
| 2018 | Furrow | 149 | 21.7 | 17 |
| ($n = 317$) | Centre pivot | 19 | 16.7 | 14.5 |
| | Sprinkler | 149 | 22 | 16 |
| Average | | | 18.9 | 14.9 |
| SD | | | 2.5 | 1.6 |

**Table A4.** Summary of the statistical tests to find whether the average seasonal actual water consumption (ET$_a$) at Xinavane estate is different. CE15

| Summary: ANOVA – single factor for ET$_a$ [mm per season] in 2015 | | | | | | |
|---|---|---|---|---|---|---|
| Groups | Count [–] | Sum* [mm per season] | Average [mm per season] | Variance [mm per season]$^2$ | | |
| Furrow | 175 | 221 623 | 1266 | 17 823 | | |
| Sprinkler | 160 | 212 857 | 1330 | 16 236 | | |
| Centre pivot | 16 | 22 621 | 1414 | 8795 | | |
| ANOVA | | | | | | |
| Source of variation | SS | df | MS | $F$ | $P$ value | $F$ critical |
| Between groups | 550 210 | 2 | 275 105 | 16.46 | $1.47 \times 10^{-7}$ | 3.022 |
| Within groups | 5 814 685 | 348 | 16 709 | | | |
| Total | 6 364 895 | 350 | | | | |

| Summary: ANOVA – single factor for $ET_a$ [mm per season] in 2016 | | | | | | |
|---|---|---|---|---|---|---|
| Groups | Count [–] | Sum* [mm per season] | Average [mm per season] | Variance [mm per season]$^2$ | | |
| Furrow | 153 | 20 1762 | 1319 | 28 102 | | |
| Sprinkler | 180 | 248 632 | 1381 | 32 201 | | |
| Centre pivot | 17 | 26 067 | 1533 | 29 346 | | |
| ANOVA | | | | | | |
| Source of variation | SS | df | MS | F | P value | F critical |
| Between groups | 852 752 | 2 | 426 376 | 14.084 | $1.315 \times 10^{-6}$ | 3.022 |
| Within groups | 10 505 019 | 347 | 30 274 | | | |
| Total | 11 357 771 | 349 | | | | |
| Summary: ANOVA – single factor for $ET_a$ [mm per season] in 2017 | | | | | | |
| Groups | Count [–] | Sum* [mm per season] | Average [mm per season] | Variance [mm per season]$^2$ | | |
| Furrow | 152 | 196 271 | 1291 | 17 828 | | |
| Sprinkler | 161 | 212 875 | 1322 | 20 093 | | |
| Centre pivot | 19 | 26 044 | 1371 | 10 756 | | |
| ANOVA | | | | | | |
| Source of variation | SS | df | MS | F | P value | F critical |
| Between groups | 147 266 | 2 | 73 633 | 3.97 | 0.020 | 3.02 |
| Within groups | 6 100 424 | 329 | 18 542 | | | |
| Total | 6 247 690 | 331 | | | | |
| Summary: ANOVA – single factor for $ET_a$ [mm per season] in 2018 | | | | | | |
| Groups | Count [–] | Sum* [mm per season] | Average [mm per season] | Variance [mm per season]$^2$ | | |
| Furrow | 149 | 187 113 | 1256 | 15 781 | | |
| Sprinkler | 149 | 193 172 | 1296 | 23 265 | | |
| Centre pivot | 19 | 27 304 | 1437 | 9258 | | |
| ANOVA | | | | | | |
| Source of variation | SS | df | MS | F | P value | F critical |
| Between groups | 585 782 | 2 | 292 891 | 15.47 | $3.91 \times 10^{-7}$ | 3.02 |
| Within groups | 5 945 377 | 314 | 18 934 | | | |
| Total | 6 531 158 | 316 | | | | |

* Sum is the product of count [–] and average [mm per season].

## Appendix B: Figures

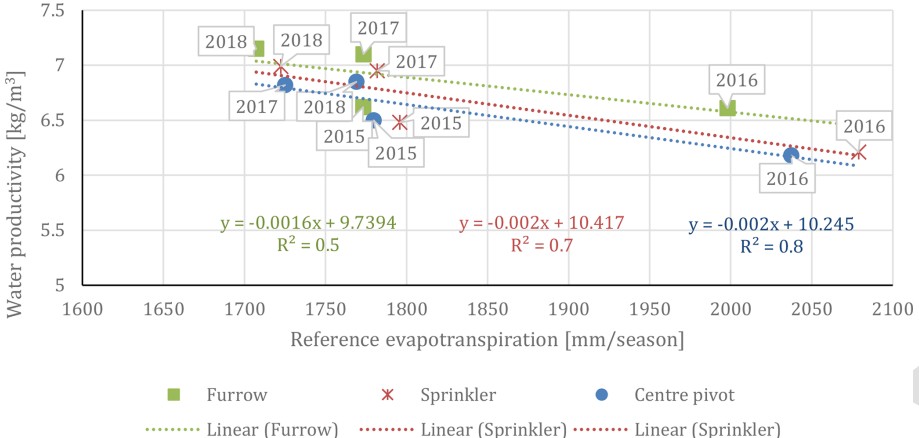

**Figure B1.** Relationship between water productivity and seasonal reference evapotranspiration at Xinavane sugarcane estate, categorized by irrigation method in 2015 to 2018.

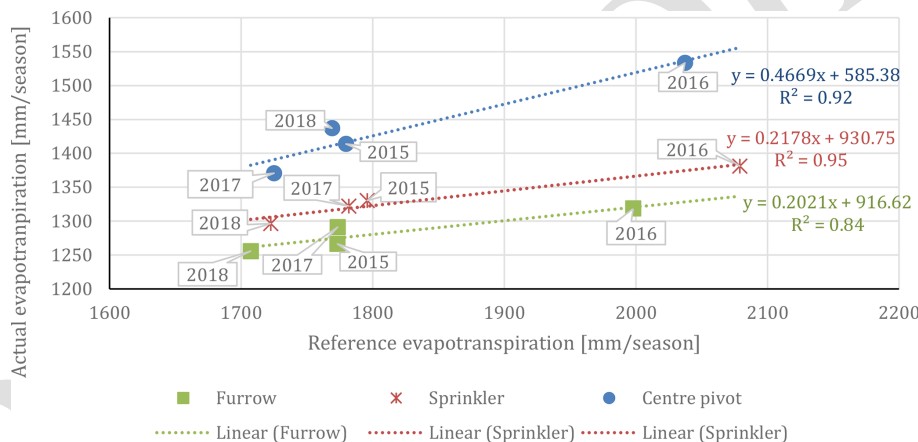

**Figure B2.** Relationship between seasonal actual evapotranspiration and reference evapotranspiration at Xinavane sugarcane estate, categorized by irrigation method in 2015 to 2018.

*Code availability.* . TS15

*Data availability.* TS16

*Supplement.* The supplement related to this article is available online at: https://doi.org/10.5194/hess-26-1-2022-supplement.

*Author contributions.* . TS17

*Competing interests.* At least one of the (co-)authors is a member of the editorial board of *Hydrology and Earth System Sciences*. The peer-review process was guided by an independent editor, and the authors also have no other competing interests to declare.

*Acknowledgements.* This study was supported by the Water Productivity Improvement in Practice (Water-PIP) project, which is supported by the Directorate-General for International Cooperation (DGIS) of the Ministry of Foreign Affairs of the Netherlands under the DGIS–IHE Delft Programmatic Cooperation (DUPC). The authors wish to thank Tongaat Hulett for their support and for sharing data of Xinavane Estate farm.

*Financial support.* This research has been supported by the NAME OF FUNDER (grant no. GRANT AGREEMENT NO). TS18

*Review statement.* This paper was edited by Dominic Mazvimavi and reviewed by two anonymous referees.

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

**Remarks from the language copy-editor**

CE1 Please note slight edit.

CE2 Is this correct as it is, or should it be "Water Management department" or "Department of Water Management"?

CE3 Please check and confirm changes.

CE4 Should "Unit" be lowercase?

CE5 Please note that since Oxford (z) spellings were almost always used in the text and in the figures, this was made consistent.

CE6 Does "-based" also apply to "water fluxes"? i.e. Should this be "are water-flux- and production-based indicators"?

CE7 Similar to above – should this be "water-delivery- and production-based"?

CE8 What do you mean by "fairly" considered?

CE9 Please check the phrase "are derived that stems from" carefully. Should this be changed to "only the data from 2014 are derived from..." or "only the data from 2014 stem from...", for example?

CE10 What do you mean by "not" practices?

CE11 What do you mean by "further" Supplement? Should this just be "see the Supplement"?

CE12 Most of the time in the text, you use "Xinavane estate". Should all cases be made uppercase like this one here? It should be made consistent.

CE13 Please check. Can "us" be deleted here?

CE14 Please confirm change from "gnomically" to "agronomically".

CE15 Please add a short sentence here to define the abbreviations ANOVA, SS, df, MS, and $F$. Thank you.

**Remarks from the typesetter**

TS1 The composition of Figs. 3–6, 10–13 and B1–B2 has been adjusted to our standards. This includes language adjustments to Figs. 4, 10, and 11.

TS2 Please confirm change throughout this paper.

TS3 Here you are referring to datasets. Please see our remark in the Data availability section regarding our standards.

TS4 Please provide last access date.

TS5 Please check placement of this row.

TS6 Is a unit missing here? Please check.

TS7 Villalobos and Fereres (2016) is missing in the reference list. Please check.

TS8 Please confirm change.

TS9 Please confirm change.

TS10 Please confirm change.

TS11 Raes et al. (2013) is missing in the reference list. Please check.

TS12 Karimi (2019) changed to Karimi et al. (2019). Please confirm.

TS13 Karimi (2019) changed to Karimi et al. (2019). Please confirm.

TS14 Karimi (2019) changed to Karimi et al. (2019). Please confirm.

TS15 Please provide a statement on how your underlying software code can be accessed. If the code is not publicly accessible, a detailed explanation of why this is the case is required. The best way to provide access to software code is by depositing it (as well as related metadata) in reliable public repositories, assigning digital object identifiers (DOIs), and properly citing code as an individual contribution. Please indicate if different software codes are deposited in different repositories or if code from a third party was used. Additionally, please provide a reference list entry including creators, title, and date of last access. If no DOI is available, assets can be linked through persistent URLs to the software code itself (not to the repositories' home page). This is not seen as best practice and the persistence of the URL must be secured.

TS16 You have referred to datasets in your text. Please provide a statement on how your underlying research data can be accessed. If the data are not publicly accessible, a detailed explanation of why this is the case is required. The best way to provide access to data is by depositing them (as well as related metadata) in reliable public data repositories, assigning digital object identifiers (DOIs), and properly citing data sets as individual contributions. Please indicate if different data sets are deposited in different repositories or if data from a third party were used. Additionally, please provide a reference list entry including creators, title, and date of last access. If no DOI is available, assets can be linked through persistent URLs to the data set itself (not to the repositories' home page). This is not seen as best practice and the persistence of the URL must be secured.

TS17 Please note that the section "Author contributions" is mandatory. Please provide the text for this section in complete sentences. Please see https://publications.copernicus.org/for_authors/obligations_for_authors.html for more information

TS18    Please note that there is funding information given in the acknowledgements, but you did not indicate any funding upon manuscript registration. Therefore, we were not able to complete the financial support statement. Please provide the missing information and double-check your acknowledgements to see whether repeated information can be removed from the acknowledgements. Thanks.

TS19    Please ensure that any data sets and software codes used in this work are properly cited in the text and included in this reference list. Thereby, please keep our reference style in mind, including creators, titles, publisher/repository, persistent identifier, and publication year. Regarding the publisher/repository, please add "[data set]" or "[code]" to the entry (e.g. Zenodo [code]).

TS20    Please provide DOI number.
TS21    Please provide page range or article number or DOI number.
TS22    Please provide full page range or DOI number.
TS23    Please provide DOI number or ISBN.
TS24    Is this a code or data set? Please check.
TS25    Please provide DOI number or URL and last access date.
TS26    Please provide DOI number or URL and last access date.
TS27    Please provide DOI number or ISBN.
TS28    Please provide volume and DOI number.
TS29    Please provide full page range or DOI number.
TS30    Please provide DOI number or URL and last access date.
TS31    Please provide DOI number or URL and last access date.
TS32    Please provide DOI number or URL and last access date.
TS33    Please provide DOI number or URL and last access date.
TS34    Please confirm change.
TS35    Please provide volume.
TS36    Please provide DOI number or URL and last access date.
TS37    Please provide full page range or DOI number.
TS38    Please provide page range or article number and DOI number.
TS39    Please provide DOI number or URL and last access date.
TS40    Please provide DOI number or URL and last access date.
TS41    Please provide DOI number or URL and last access date.
TS42    Please provide DOI number or URL and last access date.
TS43    Please provide DOI number or URL and last access date.
TS44    Please provide DOI number or URL and last access date.