# Peer review of "A Framework for Irrigation Performance Assessment Using"

_Hydrology and Earth System Sciences, 2021_

## Author Response (AR1)

We appreciate Anonymous referee #2 for the comments. Below are responses (in green) to the comments (in black).
**Note**: the line numbers are referred to the numbering in the marked-up manuscript.

*1. Does the paper address relevant scientific questions within the scope of HESS?*
**Comment:** No. I find the manuscript more suitable for an irrigation or agricultural journal and less so for the HESS journal because it presents a framework for assessing irrigation performance on sugarcane. I do not find the manuscript suitable for publication in HESS.
**Answer:** We disagree and find that this study falls squarely within the scope of HESS. First, because irrigation as a topic fits the scope. As an illustration, during the last 10 years, at least 40 papers published in HESS had "irrigation" in the title (and nearly 200 had the word "irrigation" in the abstract.[1] Second, the manuscript provides a detailed assessment of the applicability of the WaPOR database in assessing the water flux of evapotranspiration and its reliability and accuracy in relation to agricultural production. As WaPOR is promoted as a hydrological assessment tool, we are convinced that the manuscript falls within the remit of HESS.

*2. Does the paper present novel concepts, ideas, tools, or data?*
**Comment:** HESS accepts articles that clearly advances the understanding of hydrological processes and systems, and/or their role in water resources management and Earth system functioning.
The submitted manuscript does not fall under this scope.
**Answer 2**: In addition, please also see our response above (Answer 1).

*1. Are substantial conclusions reached?*
**Comment:** No. The article concludes that the framework can be used to assess performance indicators of irrigation systems. What is not reflected in the framework is how it assess the performance of irrigation systems without having integrating the amount of water delivered to each field.
**Answer 3**: We disagree with the referee who infers that all irrigation performance assessments should be based on the volume of water delivered to each field. Our explanation, which is updated in the revised manuscript [line 65-83], is as follows: Irrigation performance assessments comprise different components:
   a) water delivered and water stored in the soil (irrigation efficiency); which is not assessed here;

   b) water consumed / water applied.

Most irrigation performance assessment studies fall short of assessing actual water consumed due to lack of data (as this is notoriously difficult to measure in the field with soil moisture balances or lysimeters). Our paper, in contrast, does include this information: WaPOR provides a new RS based method to assess water consumed, enabling for the first time to assess spatial performance of irrigated crops, whereby the actual water consumed can be seen as the net outcome and result of effective rainfall and irrigation, allowing a hydrological assessment and quantification of the net water abstracted by irrigated crops. The manuscript assesses further the accuracy of this assessment through the agronomic production function (biomass-ETa).

*2. Are the scientific methods and assumptions valid and clearly outlined?*
* * *
[1] https://hess.copernicus.org/search.html?title=irrigation

**Comment:** Units for equations in the crop yield section are missing.

**Answer 4**: Thanks for the observation, we will include the units of the equations (Equation 1-3) in the revised manuscript [Line 210-218, and Line 292-293].

1. *Are the results sufficient to support the interpretations and conclusions?*

**Comment:** Statistical analysis (i.e. signifance of differences) is missing (e.g. Figure 5, Figure 6) /in error (e.g. correlation reported in not weighed by number of observations)

**Answer 5:** Thank you for the observation. The statistics that examines whether there is significance difference between the mean values of variables, e.g. seasonal actual water consumption (ETa), for the three irrigation methods is analysed (see the appendix in this response). The significance analysis showed that the mean seasonal ETa of the three irrigation methods are different. In the revised manuscript, we added the statistical analysis (Table A4 in the Appendices, from line 622 to line 628) and added text that describe the existence of significant differences between the mean seasonal actual water consumption corresponding to the three irrigation methods (Line 338-340).

**Comment:** Conclusions are made towards uniformities of irrigation systems are not born by other works. For example, uniformity (coefficient of variation of ET pixel values within a field – which the authors state that it can be used a surrogate of irrigation application distribution – 75,75,85 and 65 for centre pivot, sprinkler, drip, and furrow irrigation, respectively), cannot be derived from satellite imagery without the quantification of applied water (not water consumed). Within field variability could be due to different soil types, diseases, topography, fertilization, protection practices, etc, and not necessarily due to water. The limitations of the WAPOR data as discussed in the manuscript poses some contradictions as to the conclusions arrived at in the abstract. The paper states that the framework is useful for assessing irrigation system perfromance and variability yet it admits that these differences could be due to non-water related factors (conclusion).

**Answer 6:** The reviewer is correct that uniformity of the irrigation application is one factor affecting the uniformity of the crops. As it is also affected by for example the heterogeneous water storage capacity in the field, and hence the storage efficiency will not be uniform (leading to over-irrigation, drainage and shortages). Other factors like soil type, fertility, pest, variety etc can also affect ETa and thus uniformity. The use of water consumption (ETa) to evaluate the uniformity within an irrigated field is therefore a better indicator of the uniformity of the crops compared to evaluating water application to the fields or within fields (for which data is often not available, see also answer 3).

The manuscript also suggests the coefficient of variation could be a surrogate for the uniformity of water applied when other agricultural inputs are evenly available. This assumption is fair for an estate farm where the management is central and consistently the same level of inputs is applied. However, in the manuscript it is made clear that the 'uniformity' from ETa indicates the combined effect of all factors (water, fertility, pests, diseases, salinity) to avoid ambiguity with uniformity of irrigation water application [Line 247-252].

The main conclusion of the manuscript refers to the framework. In the abstract (line 37 and 38) it reads "We conclude that the framework is useful for assessing irrigation performance using the WaPOR dataset"; and in the conclusion section and in lines 582 – 584 in particular, it reads: "We conclude that the systematic approach demonstrated in the current study can serve as a framework to operationalize the use of WaPOR-derived data and other increasingly available RS-derived products for irrigation performance monitoring and assessment".

1. *Is the description of experiments and calculations sufficiently complete and precise to allow their reproduction by fellow scientists (traceability of results)?*

**Comment:** No, field data is not provided.

**Answer 7:** Field data of the harvested yield of more than 300 sugar cane plots during 4 consecutive years were used for validating the biomass estimates using RS. These crop yield data were provided

by the estate and will be added to our GitHub repository[2] and this link to the repository is cited in the marked-up manuscript [line 503-505]. This repository can also be used to reproduce all our calculations of the open access WaPOR data by using the protocol (framework) that we developed as part of this study which is open source and can be freely downloaded along with a step by step user guide from the same GitHub repository. The remotely sensed derived data (WaPOR) are made available through WaPOR database[3].

*2. Do the authors give proper credit to related work and clearly indicate their own new/original contribution?*

**Comment:** It seems that the methodology is derived from FAO's WAPOR manual.

**Answer 8:** We disagree with this statement, as the FAO WAPOR manual the referee refers to (and which can be downloaded from here[4]) does not have a methodology for irrigation performance assessment. The framework for irrigation performance assessment using WaPOR as presented in this paper was developed by the authors and this paper applied it to the Xinavane case study as an example of the use of the framework, and to validate the framework. The contribution of the manuscript is described from Line 85-101.

*1. Does the title clearly reflect the contents of the paper?*

**Comment:** Yes

*1. Does the abstract provide a concise and complete summary?*

**Comment:** Yes

*1. Is the overall presentation well-structured and clear?*

**Comment:** Yes

*1. Is the language fluent and precise?*

**Comment:** Almost

1. Are mathematical formulae, symbols, abbreviations, and units correctly defined and used?

**Comment:** Major equations lack units

**Answer 9:** The unit will be added (see also Answer 4)

*1. Should any parts of the paper (text, formulae, figures, tables) be clarified, reduced, combined, or eliminated?*

**Comment:** Figures need statistical significance parameters

**Answer 10**: This is agreed (see Answer 5).

*1. Are the number and quality of references appropriate?*

**Comment:** No, many references rely on reports and not peer-reviewed works.

**Answer 11:** We disagree with this statement, as only 3 out of the 52 references are non-peer-reviewed reports. These three non-peer-reviewed reports describe the database and the background of the study area but were not used to support the conclusion.

*1. Is the amount and quality of supplementary material appropriate?*

**Comment:** N/A

*Appendix*

*Table: Summary of the statistical test whether the average seasonal actual water consumption (ETa) at Xinavane estate are different*

*SUMMARY: Anova: Single Factor for ETa in 2015*
* * *
[2] https://github.com/wateraccounting/WAPORWP
[3] https://wapor.apps.fao.org/catalog/WAPOR_2/2
[4] https://www.fao.org/3/ca9894en/CA9894EN.pdf

| Groups | Count | Sum | Average | Variance |
|---|---|---|---|---|
| Furrow | 175 | 221622.6 | 1266.415 | 17823.43 |
| Sprinkler | 160 | 212857.1 | 1330.357 | 16235.72 |
| Centre pivot | 16 | 22620.74 | 1413.796 | 8795.264 |

| ANOVA | | | | | | |
|---|---|---|---|---|---|---|
| Source of Variation | SS | df | MS | F | P-value | F crit |
| Between Groups | 550209.878 | 2 | 275104.9 | 16.46461 | **1.47E-07** | 3.021669 |
| Within Groups | 5814685.29 | 348 | 16708.87 | | | |
| Total | 6364895.17 | 350 | | | | |

*SUMMARY: Anova: Single Factor for ETa in 2016*

| Groups | Count | Sum | Average | Variance |
|---|---|---|---|---|
| Furrow | 153 | 201761.5 | 1318.703 | 28101.94 |
| Sprinkler | 180 | 248631.7 | 1381.287 | 32201.07 |
| Centre pivot | 17 | 26066.95 | 1533.35 | 29345.84 |

| ANOVA | | | | | | |
|---|---|---|---|---|---|---|
| Source of Variation | SS | df | MS | F | P-value | F crit |
| Between Groups | 852751.7 | 2 | 426375.8 | 14.08397 | **1.31523E-06** | 3.021745 |
| Within Groups | 10505019 | 347 | 30273.83 | | | |
| Total | 11357771 | 349 | | | | |

*SUMMARY: Anova: Single Factor for ETa in 2017*

| Groups | Count | Sum | Average | Variance |
|---|---|---|---|---|
| Furrow | 152 | 196270.5 | 1291.253 | 17827.83 |
| Sprinkler | 161 | 212874.5 | 1322.202 | 20092.6 |
| Centre pivot | 19 | 26044.03 | 1370.738 | 10755.84 |

| ANOVA | | | | | | |
|---|---|---|---|---|---|---|
| Source of Variation | SS | df | MS | F | P-value | F crit |
| Between Groups | 147266.2 | 2 | 73633.08 | 3.971082 | **0.019764** | 3.023176 |
| Within Groups | 6100424 | 329 | 18542.32 | | | |
| Total | 6247690 | 331 | | | | |

*SUMMARY: Anova: Single Factor for ETa in 2018*

| Groups | Count | Sum | Average | Variance |
|---|---|---|---|---|
| Furrow | 149 | 187112.9 | 1255.792 | 15780.93 |
| Sprinkler | 149 | 193172.3 | 1296.458 | 23264.56 |
| Centre pivot | 19 | 27304.03 | 1437.054 | 9257.976 |

*ANOVA*

| Source of Variation | SS | df | MS | F | P-value | F crit |
|---|---|---|---|---|---|---|
| Between Groups | 585781.9 | 2 | 292890.9 | 15.46879 | **3.91E-07** | 3.024496 |
| Within Groups | 5945377 | 314 | 18934.32 | | | |
| Total | 6531158 | 316 | | | | |